# Limited and Strain-Specific Transcriptional and Growth Responses to Acquisition of a Multidrug Resistance Plasmid in Genetically Diverse *Escherichia coli* Lineages

Steven Dunn,[a] Laura Carrilero,[b] Michael Brockhurst,[b] Alan McNally[a]

[a]Institute of Microbiology and Infection, College of Medical and Dental Science, University of Birmingham, Birmingham, United Kingdom
[b]Division of Evolution and Genomic Sciences, School of Biological Sciences, University of Manchester, Manchester, United Kingdom

Steven Dunn and Laura Carrilero are co-first authors who contributed equally to this article. As the transcriptional analysis is the primary focus of this manuscript, Steven Dunn was named first given he performed the bioinformatic analysis.

**ABSTRACT** Multidrug-resistant (MDR) *Escherichia coli* strains are a major global threat to human health, wherein multidrug resistance is primarily spread by MDR plasmid acquisition. MDR plasmids are not widely distributed across the entire *E. coli* species, but instead are concentrated in a small number of clones. Here, we test if diverse *E. coli* strains vary in their ability to acquire and maintain MDR plasmids and if this relates to their transcriptional response following plasmid acquisition. We used strains from across the diversity of *E. coli* strains, including the common MDR lineage sequence type 131 (ST131) and the IncF plasmid pLL35, carrying multiple antibiotic resistance genes. Strains varied in their ability to acquire pLL35 by conjugation, but all were able to stably maintain the plasmid. The effects of pLL35 acquisition on cefotaxime resistance and growth also varied among strains, with growth responses ranging from a small decrease to a small increase in growth of the plasmid carrier relative to the parental strain. Transcriptional responses to pLL35 acquisition were limited in scale and highly strain specific. We observed transcriptional responses at the operon or regulon level—possibly due to stress responses or interactions with resident mobile genetic elements (MGEs). Subtle transcriptional responses consistent across all strains were observed affecting functions, such as anaerobic metabolism, previously shown to be under negative frequency-dependent selection in MDR *E. coli*. Overall, there was no correlation between the magnitudes of the transcriptional and growth responses across strains. Together, these data suggest that fitness costs arising from transcriptional disruption are unlikely to act as a barrier to dissemination of this MDR plasmid in *E. coli*.

**IMPORTANCE** Plasmids play a key role in bacterial evolution by transferring adaptive functions between lineages that often enable invasion of new niches, including driving the spread of antibiotic resistance genes. Fitness costs of plasmid acquisition arising from the disruption of cellular processes could limit the spread of multidrug resistance plasmids. However, the impacts of plasmid acquisition are typically measured in lab-adapted strains rather than natural isolates, which act as reservoirs for the maintenance and transmission of plasmids to clinically relevant strains. Using a clinical multidrug resistance plasmid and a diverse collection of *E. coli* strains isolated from clinical infections and natural environments, we show that plasmid acquisition had only limited and highly strain-specific effects on bacterial growth and transcription under laboratory conditions. These findings suggest that fitness costs arising from transcriptional disruption are unlikely to act as a barrier to transmission of this plasmid in natural populations of *E. coli*.

**KEYWORDS** *Escherichia coli*, transcriptomics, antimicrobial resistance, plasmids

Address correspondence to Alan McNally, a.mcnally.1@bham.ac.uk.

Multidrug-resistant (MDR) *Escherichia coli* strains present a global public health risk and are listed by the World Health Organization as a priority pathogen. The incidence of MDR *E. coli* strains as etiological agents of human disease has steadily increased since the turn of the century (1). Initially this was due to the emergence of *E. coli* clones carrying plasmids containing extended-spectrum β-lactamase (ESBL) genes conferring resistance to third-generation cephalosporins (1). This emergence mirrored the rise in the incidence of *E. coli* as the causative agent of bloodstream infections worldwide, primarily due to the rapid global dissemination of MDR clones (1, 2). This was followed by the emergence of clones carrying plasmids carrying carbapenemase enzyme genes, conferring resistance to all antimicrobial classes, with the exception of colistin (3, 4).

The emergence of MDR *E. coli* has not occurred evenly across this species. Rather MDR plasmid carriage is concentrated in a number of clones associated with extraintestinal infections, while it is rarely seen in clones causing intestinal infectious disease or in exclusively commensal lineages (5). ESBL plasmid carriage is most commonly seen in low-diversity clones of lineages such as sequence type 131 (ST131), ST648, and ST410 (5), with ST131 representing the most common cause of MDR *E. coli* bloodstream and urine infections in the developed world (2). Carriage of carbapenemase-encoding plasmids is also concentrated in low-diversity clones of lineages such as ST167 and ST410, belonging to the phylogroup A clade of *E. coli*, which are generally devoid of most common *E. coli* virulence factors (4, 6, 7).

Comparison of the genomes of MDR plasmid-carrying clones with the lineages that those clones emerged from shows striking similarity in key steps in their evolution. All show rapid clonal expansion of MDR plasmid-carrying strains, which become globally distributed in a matter of years (6–8). The MDR clones also carry clone-specific alleles of key genes encoding traits involved in human colonization, such as adhesins and iron acquisition (6, 7, 9). Comprehensive analysis of ST131 MDR clade C showed it differed from the drug-susceptible clades A and B of the lineage in a number of unique alleles of genes involved in colonization as well as anaerobic metabolism genes (9). MDR clones also contain unique intergenic sequence alleles, which correlate with plasmids carried by strains (6, 7, 10).

As well as the biosynthetic burden associated with replicating, transcribing, and translating the new genetic material, plasmid acquisition often disrupts normal cellular function. For example, large-scale changes to regulation of chromosomal genes have been observed following plasmid acquisition in a range of bacterial hosts (11–13), which can be negated by compensatory mutations to regulators. The shared genetic traits of MDR clones of *E. coli*, including regulatory sequences together with the uneven distribution of MDR plasmids, suggest that some lineages of *E. coli* may be more tolerant to potential disruptions from the acquisition and stable integration of MDR plasmids than others (5, 10). However, data comparing the transcriptional and phenotypic responses of diverse *E. coli* strains to MDR plasmid acquisition are lacking.

We tested the transcriptional response to acquisition of pLL35, an ESBL plasmid encoding CTX-M-15 and TEM-112, in eight genetically diverse *E. coli* strains, including environmental *E. coli* isolates from lineages in which MDR plasmids have never been reported and strains from clades A, B, and C of *E. coli* ST131, wherein clade C is most frequently associated with MDR plasmid acquisition. Strains varied in the rate of pLL35 acquisition by conjugation from *Klebsiella pneumoniae* and the degree of cefotaxime resistance conferred by the plasmid but not in the stability of the plasmid once acquired. pLL35 carriers showed variations in growth relative to plasmid-free cells ranging from impaired to enhanced relative growth of plasmid carriers. Plasmid transcription did not vary significantly among host strains, but we observed strain-specific differences in chromosomal gene expression caused by plasmid acquisition. We observed no correlation between the degree of transcriptional disruption caused by plasmid acquisition and the relative growth of pLL35 carriers.

mSystems®

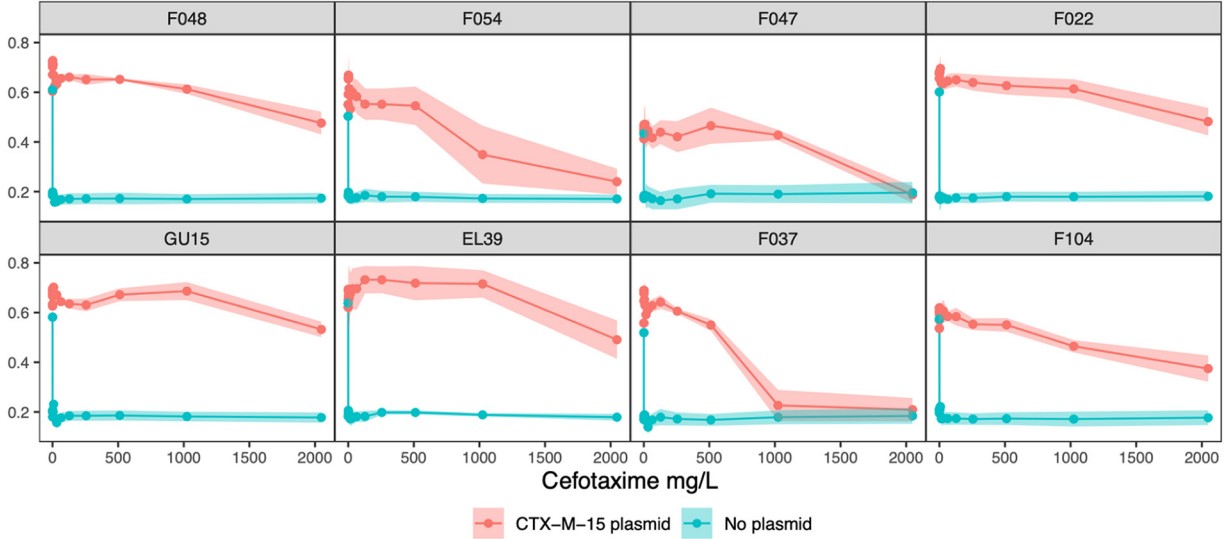

**FIG 1** The level of resistance to cefotaxime conferred by the pLL35 varied by strain. Plots are facetted by *E. coli* strain. Solid lines show the mean ($n = 3$) $\pm$ standard error of relative growth measured using the optical density at 600 nm (OD$_{600}$) with (red) or without (blue) pLL35 across gradients of increasing cefotaxime concentration. Horizontal dashed lines indicate a relative growth value of 0.5 (i.e., the MIC$_{50}$).

## RESULTS

***E. coli* strains varied in conjugational uptake of pLL35.** For most strains, the conjugation rate from *K. pneumoniae* was higher in static than in shaken cultures. Indeed, for several strains we detected no transconjugants from shaken cultures in any of the replicates (e.g., F022 and F047: ST131 clades A and C, respectively), and only three strains consistently acquired the plasmid under both shaken and static conditions (i.e., F037, F048, and F054: all ST131 clade B/C) (see Fig. S1 in the supplemental material). Due to the large number of missing replicates in shaken cultures, we were only able to analyze variation across all strains in the conjugation rates estimated from static cultures. In static cultures, strains varied in their ability to acquire the plasmid by conjugation from *K. pneumoniae* (analysis of variance [ANOVA], strain effect, $F_{7, 36} = 19.23$, $P = 2.06e-10$) (see Fig. S2 in the supplemental material). Once acquired, however, the plasmid was stably maintained over time by all strains (Wilcoxon test comparing population density averaged over time on media with or without cefotaxime, all strains, $P > 0.05$) (see Fig. S3 in the supplemental material).

**Strain-specific effects of pLL35 on resistance and bacterial growth kinetics.** Plasmid acquisition increased resistance to cefotaxime, but the level of resistance conferred by the plasmid varied by strain (ANOVA, strain by plasmid interaction, $F_{7, 32} = 2.968$, $P = 0.000219$) (Fig. 1). Specifically, in strains F037, F047, and F054, the plasmid provided lower levels of cefotaxime resistance than in the other strains bearing plasmids.

Parental strains varied in their growth parameters (lag time, ANOVA, $F_{7, 32} = 5.358$, $P = 0.000394$; maximum growth rate, ANOVA, $F_{7, 32} = 6.3$, $P = 0.000108$; saturation density, ANOVA, $F_{7, 32} = 18.48$, $P = 0.00000000136$; integral of the growth curve, Kruskal-Wallis $\chi^2_7 = 31.759$, $P = 0.000045$). To control for this variation in baseline growth among the strains, we normalized growth parameters per plasmid-carrying strain by its corresponding parental strain. This gave estimates of relative growth parameters and thus of the impact of plasmid acquisition upon the growth of each strain (i.e., a value of 1 would indicate no effect on growth of plasmid acquisition) (Fig. 2). Strains varied in their relative maximum growth rate (ANOVA, $F_{7, 32} = 5.742$, $P = 0.00023$), relative saturation density (Kruskal-Wallis $\chi^2_7 = 24.102$, $P = 0.001093$), and relative integral of the growth curve (ANOVA, $F_{7, 32} = 8.998$, $P = 0.00000424$), but not in their relative lag time (Kruskal-Wallis $\chi^2_7 = 10.264$, $P = 0.1741$). The clearest impacts of plasmid acquisition on

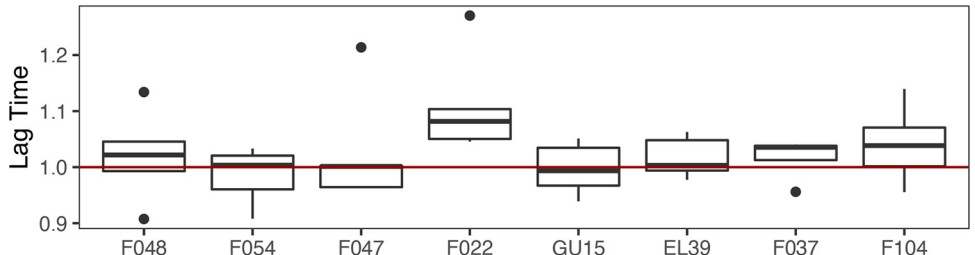

Lag Time plasmid bearers / Lag Time parental strain

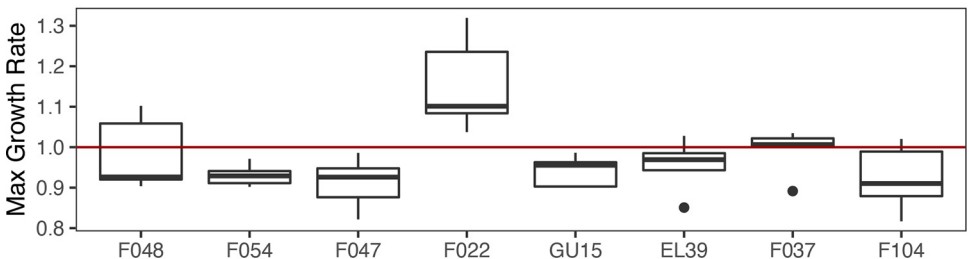

Max Growth Rate plasmid bearers / Max Growth Rate parental strain

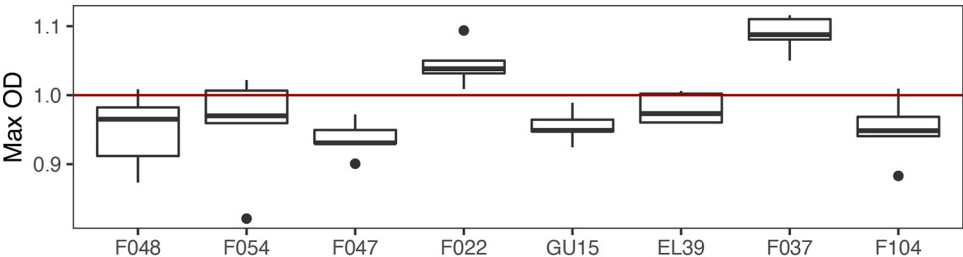

MaxOD plasmid bearers / MaxOD parental strain

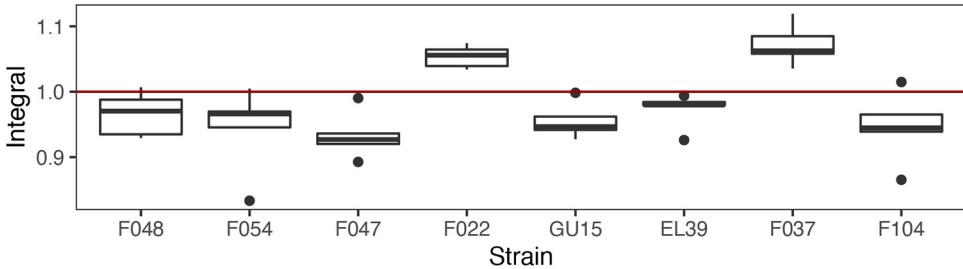

Integral plasmid bearers / Integral parental strain

**FIG 2** The effect of acquiring pLL35 on bacterial growth kinetics varied by strain. Panels show the responses of the following growth parameters to plasmid acquisition: (A) lag time, (B) maximum growth rate, (C) maximum density, and (D) integral (i.e., area under the growth curve). Boxes show normalized values (plasmid-carrying strain value divided by parental strain value) for each strain ($n = 5$) indicating the performance of plasmid carriers relative to their parental strain, where a value of 1 represents equal performance. The lower hinge of the box denotes the 25th percentile, the upper hinge denotes the 75th percentile, and the line within the box indicates the median. Upper whiskers extend to the highest value no further than 1.5 times the interquartile range from the hinge. Lower whiskers extend to the smallest value no further than 1.5 times the interquartile range from the hinge. Points indicate outliers beyond the whiskers.

growth were apparent for the relative integral of the growth curve (Fig. 2), which is a useful measure of the overall effect of plasmid acquisition on growth (14): the integral of the growth curve was reduced by plasmid carriage in the strains GU15 and F047 (one-sample $t$ test of relative integral against 1, GU15, $t = 3.6933$, df $= 4$, $P = 0.021$; F047, $t = 4.1762$, $P = 0.014$) but was increased in the strains F022 and F037 (one-sample $t$ test of relative integral against 1, F022, $t = 7.0987$, df $= 4$, $P = 0.0021$; F037, $t = 5.0836$,

$P$ = 0.0071). Thus, acquisition of the ESBL plasmid had variable effects upon growth across the strains, causing both increased and decreased growth, while having a negligible impact upon the growth of half of the strains tested.

Following the masking of any variants that occurred in the ancestral or control sequence data, ESBL plasmid carriers contained very few single nucleotide polymorphisms (SNPs), with the majority of isolates containing no SNPs at all ($n = 6$ out of 8). The position of detected SNPs was determined, but no clear evidence of parallelism could be established (see Table S2 in the supplemental material). To confirm this limited genomic impact further, we sequenced three additional independently constructed transconjugant replicates. This also revealed a small number of SNPs (0 to 2 SNPs per replicate), with the majority of sequences containing 0 variants ($n = 16$ out of 24).

**Strain-specific transcriptional responses to acquisition of pLL35.** We next compared the transcriptomes of the pLL35-carrying transconjugants with their parental strain to determine the effect of plasmid acquisition on host gene expression. Combining the results from differential gene expression analyses of three independent biological replicates showed very little significant transcriptional response in any strain, with the number of significantly (>2-log fold change [FC]) differentially expressed genes ranging from 22 to zero at a false-discovery rate (FDR) $P$ value of <0.05 and 31 to zero at an FDR $P$ value of <0.1 (Fig. S3). Volcano plots for the transcriptional impact of plasmid acquisition per strain show highly strain-specific responses to acquisition of pLL35, both in terms of the level of transcriptional response and the genes that were differentially expressed (Fig. 3). Strains ELU39, F104 and F037 showed no significant changes in gene expression upon acquisition of the plasmid (at an FDR $P$ value of <0.05 or <0.1), strains GU15, F054, and F048 had fewer than 5 genes significantly differentially expressed, and strains F022 and F047 showed between 10 and 22 genes significantly differentially expressed (at an FDR $P$ value of <0.05) (see Fig. S4 in the supplemental material). There was no correlation between magnitude of transcriptional response and the growth responses observed in the strains ($r = -0.6162$, $P = 0.1038$).

**Functions whose expression was affected by pLL35 varied between strains.** We observed differences in the functions whose expression was significantly affected by pLL35 acquisition between the strains. In F022 plasmid carriers, we observed upregulation of the entire class 1, 2, and 3 flagellar biosynthesis regulons relative to the plasmid-free parental strain (15). Further inspection of the genome sequences of the parental and transconjugant F022 strains revealed that this occurred due to an insertion of an IS*1* family IS element in *lrhA*, the negative regulator of *flhDC* (Fig. 4).

In F047, pLL35 caused upregulation of a variety of chromosomal genes, including those involved in various stress responses, such as *cpxP* (envelope stress response), *deaD* (low-temperature response), *ibpAB* (heat and oxidative stress), and *soxS* (superoxide stress master regulator). This is mirrored in several differentially expressed genes just below the 2-log fold change (FC) significance threshold, such as *dnaJ*, *degP*, and *osmY* (FCs of 1.81, 1.64, and 1.59), which are all involved in stress response. The presence of the plasmid also led to upregulation of *marR*, the repressor of the *mar* antibiotic resistance and oxidative stress response regulon, though *marA* expression was not significantly affected (FC of 1.0, FDR of 0.57). Other functions upregulated by plasmid acquisition included metabolic transport genes (*mgtA* and *pstS*) and anaerobic metabolism genes (*glpD*).

In F048, several colocated hypothetical genes were upregulated in plasmid carriers: 3 of these were significantly upregulated, while the remainder were slightly below the FDR significance threshold. This region was further characterized with Prophage Hunter, showing a putative ~60-kb prophage. This prophage contains 78 genes and shows 94% identity and 28% coverage with prophage CUS-3. No evidence of phage mobilization was detected by structural variant analysis, and the read depth at this region (normalized to housekeeping genes) was consistent across both the parental and transconjugant read sets, suggesting that the observed upregulation was not due to the presence of additional prophage copies.

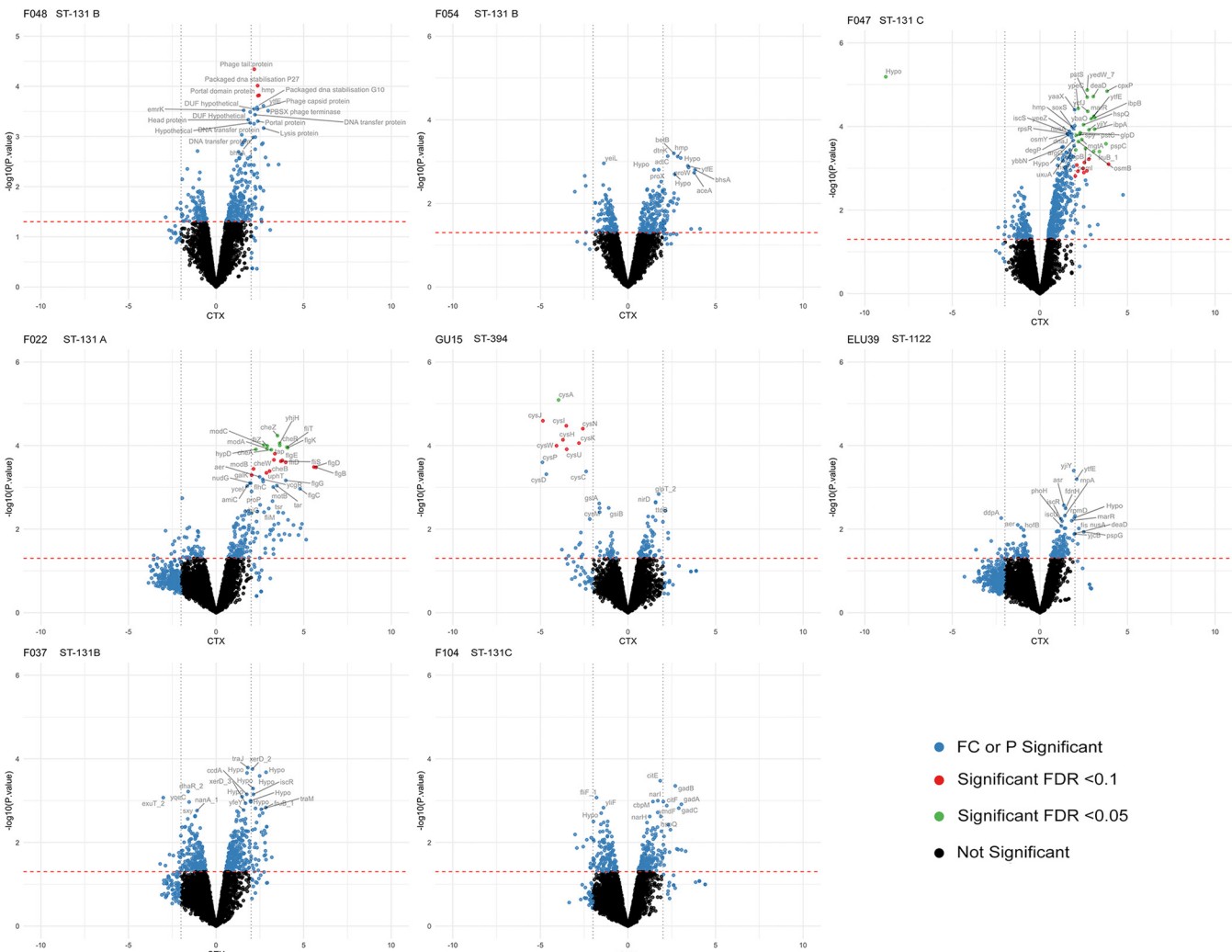

**FIG 3** Transcriptional responses to acquiring pLL35 varied by strain. Log$_2$ fold change (CTX) of differentially expression data and their statistical significance ($-\log_{10}$ of the $P$ value). While 4 of the 8 isolates showed no significant transcriptional differences, other isolates showed patterns of differential expression in discrete operons or in genes under the control of a common regulator. The transcriptomic effects observed in this host range are determined by strain, rather than host genetic background. Blue indicates significant fold change ($>2$) or significant $P$ value ($<0.05$), red indicates the FDR was also significant at a threshold of $<0.10$, green indicates the FDR was also significant at a threshold of $<0.05$, and black indicates not significant.

In GU15, pLL35 led to downregulation of the entire sulfur biosynthesis and transport operons (*cysA* to *cysW*). Significant downregulation was observed in *cysA* at an FDR of $>0.05$ and genes *cysHIJKNWU* at an FDR of $>0.1$. The rest of the operon genes were expressed at ranges slightly below the FDR significance threshold.

**Consistent pattern of low-level transcriptional response to pLL35.** Some genes known to be in operons or regulons did not pass the significance threshold in our FDR analysis, despite the rest of the operon or regulon doing so. This is due to our use of amalgamated data from three independent biological replicates and strict significance thresholds. We decided to further analyze the expression of all core genes (see Fig. S5 in the supplemental material) using a principal-component analysis to detect genes that were differentially expressed across our host range (see Fig. S6 in the supplemental material). Analysis of this set of genes showed some consistent patterns of differential expression across all strains. A common pattern of upregulation in response to acquisition of pLL35 was observed for the *cit* operon, *his* operon, the *hya*, *hyb*, and *hyc* operon, and the *nar* and *ttd* operons and *gad* loci (Fig. 5). Conversely, there was a common pattern of downregulation in response to acquisition of pLL35 of the *csg*, *gat*, *waa*, *yad*, and *yih* operons across all strains. Functional enrichment analysis of genes in

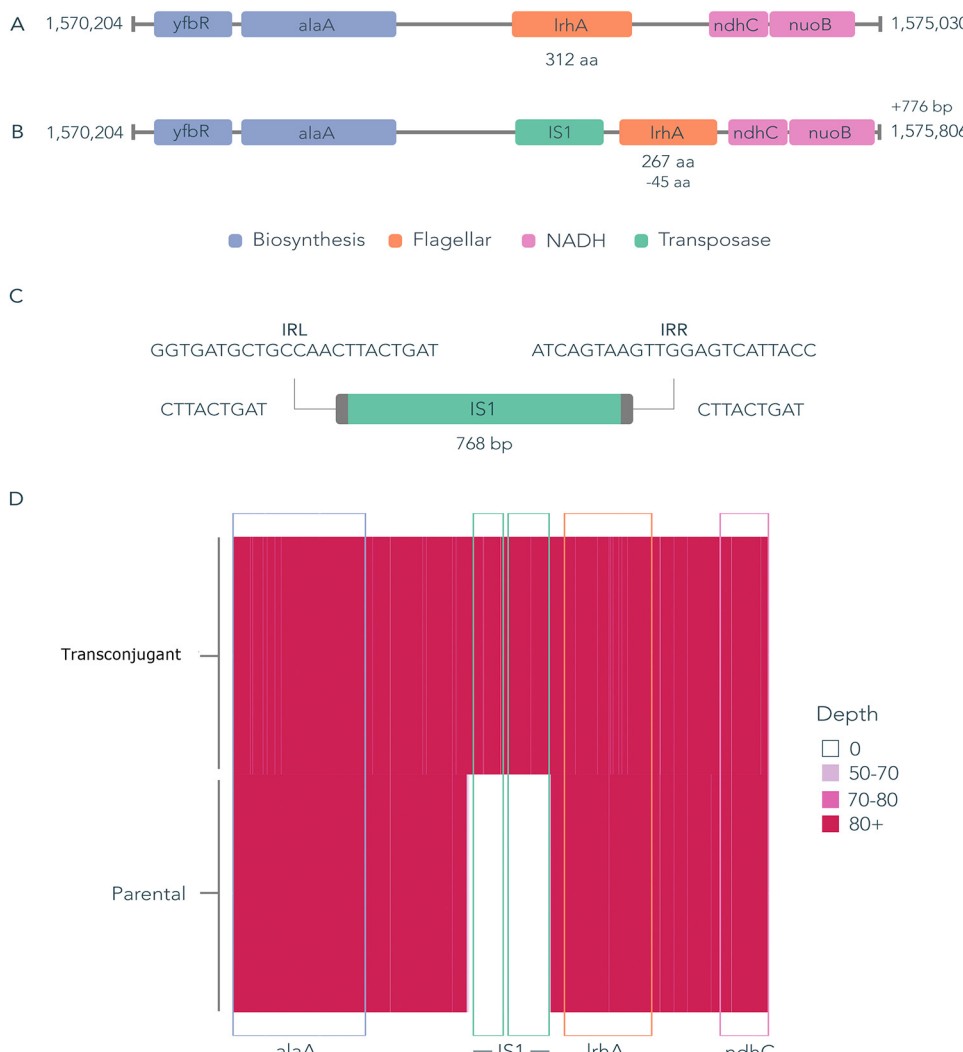

**FIG 4** The transcriptional response of strain F022 was likely caused by a chromosomal insertion of an IS element. (A) Parental genome sequence with fully intact *lrhA* gene. (B) Transconjugant genome sequence with an IS1 family transposase sequence causing a truncation to *lrhA*. (C) IS1 family transposase with left and right inverted repeat sequences (IRL and IRR, respectively) and a 9-bp target site duplication. This particular IS1 occurs 7 times in the parental strain and was inserted one additional time upon plasmid conjugation. Querying the ISfinder database shows that this IS1 shares 97% sequence identity with IS1 R, B, and D. This element carries two ORFs: *insA* and *insB*. (D) Depth of long reads uniquely mapped to the transconjugant assembly. The minimum depth of reads that were successfully mapped to this region is 50, and in the parental isolate, there are 0 reads that map to the IS1 transposase.

the 5th and 95th percentiles of these differentially expressed loci confirmed this consistent fine-scale transcriptional response to the plasmid in genes associated with the cell wall, signal transduction, cell motility, energy production and conversion, and carbohydrate transport and metabolism (Fig. 5).

## DISCUSSION

Evidence from experimental evolution studies has provided us with a detailed picture of the impact that acquisition of plasmids and their stable integration into a host cells genetic inventory have on cell fitness (16). Much of this fitness impact is driven by changes in transcription in the acquiring cell: both the need to transcribe genes on the plasmid, but also global effects on host cell transcription to offset the impact of carrying the plasmid (11, 13, 17). There are very few of these studies examining the impact of acquisition of multidrug resistance plasmids on cells from genetically diverse strains

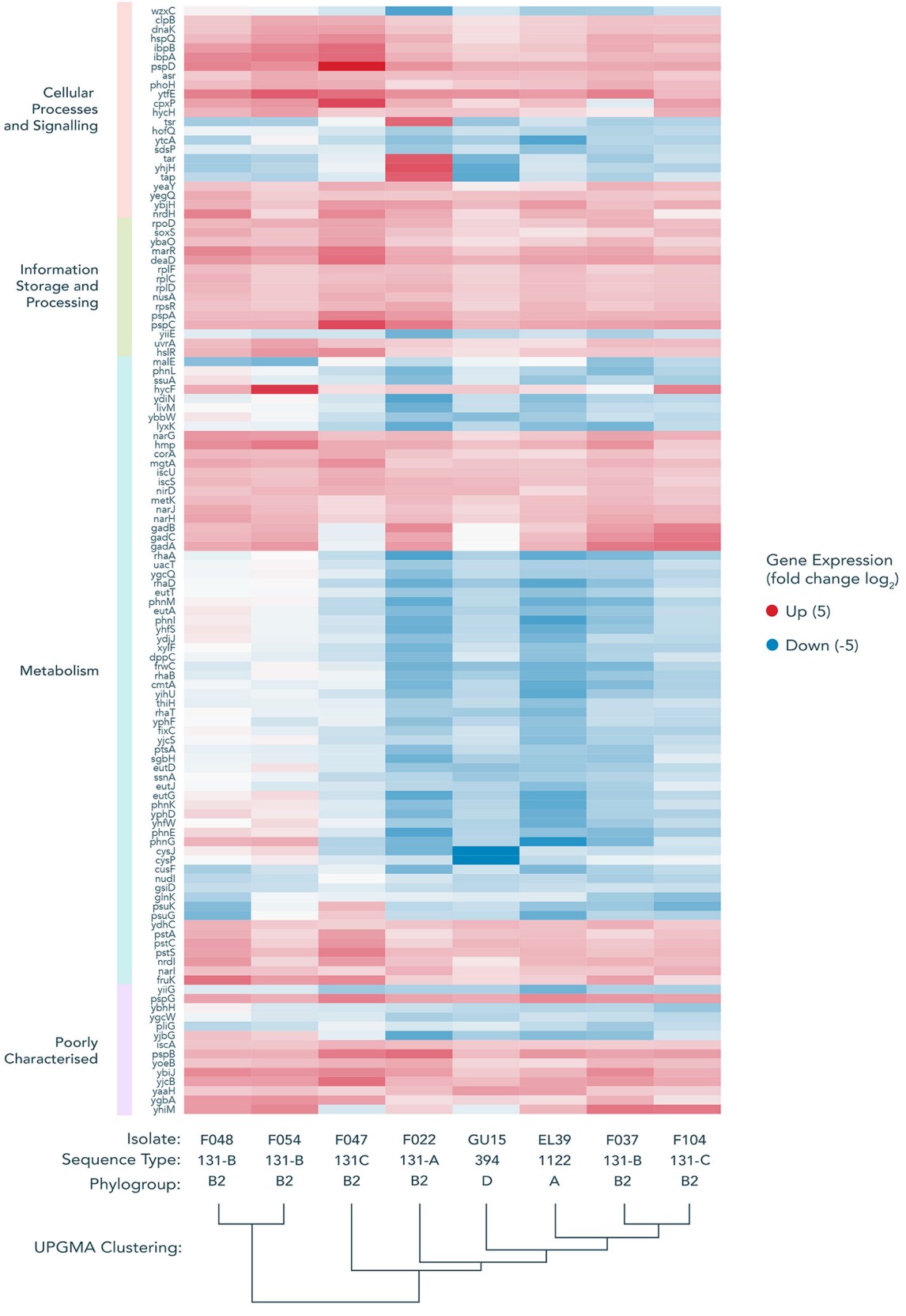

**FIG 5** Transcriptional change of genes from the extremes of the PCA distribution show some common signatures of differential expression (e.g., *cit* operon, *his* operon, the *hya*, *hyb*, and *hyc* operon, and the *nar*, *gad*, and *ttd* operons). Genes were extracted from the 5th and 95th percentiles of the PCA distribution from Fig. S6 (i.e., genes extraneous to the central distribution). These genes were assigned to COG functional categories, with each category ordered via UPGMA clustering of the expression data.

(17), with most evidence of adaptations that occur as a result of MDR plasmid acquisition stemming from large comparative population genomics studies (7, 9, 10). Here, we address this by examining the impact of acquisition of an MDR plasmid on *E. coli* strains from a variety of genetic backgrounds, ranging from environmental lineages with no reported MDR plasmid carriage to MDR-plasmid-free clinical isolates from the ST131 lineage most commonly associated with multidrug resistance in clinical settings.

Acquisition of the ESBL plasmid pLL35 varied among strains, with some strains unable to successfully conjugate under shaking conditions. All strains stably maintained the plasmid once they had acquired it, likely due to the presence of two toxin/antitoxin systems on pLL35. Plasmid acquisition had variable effects on growth between strains. Notably, plasmid acquisition was not costly in terms of relative growth for all the strains, with increased growth of plasmid carriers relative to their parental strains observed in two strains. This is surprising given that plasmid acquisition has been shown to be associated with fitness costs across a diversity of plasmid-host interactions, although variation in the magnitude of the cost has been described (18–21). However, our data are consistent with those of another recent study that tested the fitness effect of a given plasmid across the range of genetic backgrounds present in a host species. Like ours, this study revealed diverse fitness impacts ranging across a continuum from costly to beneficial (22). These data highlight that the fitness effects of a plasmid can be highly strain specific, and thus are likely to arise from specific genetic interactions rather than the generic biosynthetic costs of plasmid maintenance. Interestingly, effects on growth of plasmid acquisition were uncorrelated with the extent of changes in gene expression across the strains, suggesting that greater plasmid-mediated gene dysregulation does not necessarily translate to larger fitness costs.

Strains varied in the level of cefotaxime resistance provided by the ESBL plasmid. This suggests epistasis between the plasmid ESBL gene and chromosomal loci that vary among strains. Through comparison of the strain genomes, we could not identify any clear differences in chromosomal gene content among strains in terms of known resistance determinants to explain the observed variation in cefotaxime resistance. For example, all strains encode the same set of standard efflux pumps, with the exception of F047 and F048, which also contain *tetA*. Yet F047 and F048 differ markedly to each other in their cefotaxime resistance response, suggesting that TetA does not explain the variable resistance response. All strains carry a variant of $bla_{EC}$. In addition, F022 and F047 carry $bla_{TEM-1}$ but vary in their cefotaxime resistance responses, suggesting that $bla_{TEM-1}$ does not explain the variable resistance response. This is perhaps unsurprising as the prediction of a strain's resistance phenotype from gene content alone is notoriously inaccurate (23).

Comparison of the transcriptomes of plasmid carriers with their parental strain revealed highly strain-specific effects of plasmid acquisition on the expression of chromosomal genes. The number of genes whose transcription was affected by the plasmid was small in all strains, ranging from 0 to just 22 genes (at an FDR of <0.05). These data stand in contrast to other studies where the expression levels of hundreds of chromosomal genes are affected by plasmid acquisition (11, 13, 24, 25). Significant transcriptional effects of ESBL plasmid acquisition were focused in discrete operons or regulons. For example, in GU15, we observed downregulation of the sulfur biosynthesis and transport operons (*cysA* to *cysW*). In F048 plasmid carriers, flavohemoglobin (Hmp), which is responsible for resistance to nitrosative stress (26), was upregulated following plasmid acquisition. In F047, the transcriptional impact of plasmid acquisition was more widespread, affecting more diverse functions, but consistent with F048, most of these were related to stress responses. Upregulation was observed in *marR*, which did not extend to *marA*; *marA* has been demonstrated to have a short half-life (3 min) and is quickly depleted when the environmental stress is removed (27). Increased expression was also observed in heat shock proteins IbpA/B. The function of IbpA/B extends beyond heat stress; previous studies have shown that increased *ibpA/B* expression resulted in the overproduction of a β-lactamase precursor, potentially through

IbpA/B binding to the precursor protein, preventing subsequent processing (28). This could explain the lowered cefotaxime MIC observed, though the exact cause of *ibpA/B* upregulation is unclear. All of these factors indicate that F047 exhibited a strong stress response to acquisition of the plasmid. Plasmids are known to elicit stress responses in their host cells (12): for example, through the conjugation-mediated cell envelope stress in *E. coli* (29) or due to single-stranded DNA activating the SOS response (30). Additionally, pLL35 carries DNA polymerase V genes *umuDC*, the chromosomal homologs of which are part of the SOS response regulon (31, 32). Intriguingly, a plasmid-carried *umuD* gene has previously been shown to regulate the SOS response in *Pseudomonas aeruginosa* (33), suggesting that plasmids may directly manipulate expression of the bacterial SOS response. It is hypothesized the production of a conjugation apparatus may lead to misfolded proteins; F047 failed to conjugate under shaking conditions. This may be due to an increase in the concentration of damaged or misfolded proteins beyond that which would be mitigated by increased expression of stress response proteins (e.g., DegP and CpxP).

In two of our strains, there is evidence of a transcriptional response possibly driven by the relationship between mobile genetic elements (MGEs). In F022, the upregulation of flagellar and chemotaxis genes was explained by a 768-bp insertion of an IS*1* family element carrying both *insA* and *insB* in the *lrhA* regulator. LrhA belongs to the LysR family and binds to and negatively regulates expression of the *flhDC* master regulator. Truncation of *lrhA* is therefore likely to have prevented negative regulation of *flhDC*, leading to uncontrolled expression of the flagellar and chemotaxis operons. Genomic comparison of IS elements within the F022 genome revealed 7 identical IS*1* sequences in the parental genome, with 4 found on the chromosome and 3 on a canonical plasmid. Upon plasmid acquisition, this element inserted one additional time. This appeared to be a random event, as a similar insertion could not be detected in the short-read data of replicate transconjugants. In F048, a set of colocated upregulated genes were associated with a chromosomal prophage. Many prophages respond to host stress responses, which can be induced by plasmid acquisition and thus may explain the upregulation of prophage gene expression observed here. Prophages have also been shown to excise and replicate under stress conditions, but we did not detect any excision or genomic amplification of the phage region.

Besides those genes whose expression was significantly altered by ESBL plasmid acquisition (i.e., that met the stringent significance threshold), we also observed a subtle but consistent transcriptional response to plasmid acquisition among all genes with a >2-log fold change in expression. Upon acquisition of a plasmid, the most intuitive scenario would be a fall in chromosomal transcription as transcriptional machinery is sequestered at plasmid promoters (5). Accordingly, across all strains, we observed reduction in transcription of *csg* genes encoding curli fimbriae and genes involved in cell wall and outer membrane production, including the *waa* lipopolysaccharide (LPS) core genes. These genes are involved in biosynthesis of energetically costly structures in the cell, and their repression is consistent with offsetting energetic costs of plasmid maintenance. Conversely, we observed consistently increased transcription of *hya*, *hyb*, and *hyc* genes encoding the hydrogenase-1 and -3 complexes, the *nar* gene encoding nitrate reductase, the *ttr* gene encoding tartrate dehydratase, and the *gad* glutamate decarboxylase operon. All of these genes are involved in various aspects of anaerobic metabolism, which is known to be important for colonizing the mammalian gut. Moreover, some of these genes exhibit negative frequency-dependent selection in the MDR clade C of *E. coli* ST131, which may reflect selection for enhanced intestinal colonization (9). The observation of broad-scale, subtle changes to chromosomal gene expression caused by an MDR plasmid that are consistent across diverse bacterial lineages warrants further investigation. Their scale is suggestive of a role for plasmid-encoded regulatory elements, such as small RNAs (34), with the potential for genome-wide effects.

**Conclusion.** We observed strain-specific but limited effects of acquisition of an ESBL plasmid across diverse *E. coli* lineages. The transcriptional response to plasmid

acquisition was limited to differential expression of small numbers of genes within discrete operons or regulons whose identity varied between strains. More subtle but consistent effects of plasmid acquisition on global transcription were observed, affecting a range of cellular processes. Relative growth and cefotaxime resistance of ESBL plasmid carriers varied between strains. Overall, our findings suggest that the effects of MDR plasmid acquisition upon the host cell arise from specific genetic interactions that are likely to be difficult to predict *a priori* and that fitness costs are unlikely to act as a barrier to transmission of this MDR plasmid in natural populations of *E. coli*.

## MATERIALS AND METHODS

**Bacterial strains and plasmids.** A total of 8 *E. coli* strains were selected for use in this study, representing sequence type 131 (ST131: clades A, B, and C), ST394, and ST1122 (see Table S1 in the supplemental material). All strains were screened to ensure that they did not contain any existing MDR plasmids. This study used plasmid donor strain LL35 (a *Klebsiella pneumoniae* isolate belonging to ST45), which contains pLL35, a 106-kb IncFII(K)-9 plasmid with a complete conjugative transfer machinery, and a complex antibiotic resistance region (Fig. S1). The resistance region in pLL35 is comprised of multiple translocatable genetic elements. It contains several complete antibiotic resistance genes, conferring resistance to cephalosporins and $\beta$-lactams ($bla_{CTX-M-15}$ and $bla_{TEM-112}$), aminoglycosides (*aacA4*, *aacC2*, and *aadA1*) and quinolones (*qnrS1*). This region also contains OXA-9; however, this gene is truncated due to a premature stop codon. The antibiotic resistance region is potentially mobilizable, due to an ISEcp1 insertion sequence. pLL35 also encodes two separate toxin/antitoxin systems (*higAB* and *ccdAB*).

**Plasmid conjugation.** Twelve independent conjugation assays were performed for each strain following an endpoint method (35). A single colony from overnight growth on nutrient agar was inoculated into 5 ml of nutrient broth (NB; Oxoid, United Kingdom). This was incubated at 37°C for 2 h with shaking (180 rpm). Cultures were mixed at a ratio of 1:3 donor to recipient, and 50 $\mu$l was used to inoculate 6 ml of brain heart infusion (BHI). Six replicates were incubated as static cultures at 37°C for 24 h, and six replicates were incubated as shaken cultures at 37°C for 24 h at 180 rpm. Conjugation rate data were $\log_{10}$ transformed for statistical analysis.

The conjugation mixture was plated onto UTI Chromagar (Sigma-Aldrich, United Kingdom) supplemented with 4 $\mu$g/ml of cefotaxime and incubated at 37°C overnight. Colonies that produced a pink color indicative of *E. coli* were further subcultured onto UTI Chromagar with 4 $\mu$g/ml of cefotaxime in order to check the resistance profile and to ensure there was sufficient pure growth to store for subsequent use. One of these replicates was used to quantify differential gene expression and is subsequently referred to as the transconjugant. Whole-genome sequencing (WGS) data were also generated for three additional replicates from the conjugation assays, which are referred to as the transconjugant replicates.

For the generation of conjugated strains, bacteria were subcultured a total of 5 times. In order to account for any confounding effects of adaptation to the lab conditions, and to control for any variation generated by variables extraneous to the plasmid conjugation, the parental recipient strains were also run through the conjugation protocol with a plasmid-free *Klebsiella* strain, Ecl8 (17). These triplicate samples were also sequenced and are referred to as the control replicates.

**Genome sequencing.** Whole-genome sequencing was performed on the ancestral, transconjugant, control replicate, and transconjugant replicate strains. The ancestral and transconjugant lines were sequenced by both Illumina- and Oxford Nanopore-based technologies. Illumina sequencing was provided by MicrobesNG (http://www.microbesng.com), and DNA was extracted from an overnight bacterial suspension by MicrobesNG, using an automated SPRI bead-based workflow. Illumina genome sequence reads were assessed for quality using FastQC (v.0.11.9) and subsequently trimmed using Trimmomatic (v.0.3) with a sliding window quality of Q15 and length of 20 bp. Kraken (v.2) was used to confirm species ID and check for potential contaminants.

Long-read sequencing was performed on unsheared DNA extracted from an overnight bacterial suspension using a phenol-chloroform method (36). DNA was quantified using the Qubit 4.0 and a broad-range double-stranded DNA (dsDNA) kit (ThermoFisher, United Kingdom). Libraries were prepared using the SQK-LSK109 sequencing kit and EXP-NBD104 expansion set, following the manufacturer's protocol. The libraries were then sequenced on a MinION rev 4.1D using an R9.1 flow cell over 48 h (Oxford Nanopore Technologies, United Kingdom).

MinION data were base called using GPU-accelerated Guppy (v.3.1.5 + 781ed57) in high-accuracy mode. Adapters were confirmed and removed using Porechop (v.0.2.3_seqan2.1.1). Reads that had differential demultiplexing via Guppy and Porechop were discarded, leaving only reads for which both programs had reached a consensus. Chimeric reads were discarded using Unicycler's Scrub module (v.0.4.7). Finally, reads were filtered using FiltLong (v.0.2.0), with parameters based on read length and quality distributions generated by NanoPlot (v.1.24.0), removing relatively short or low-quality reads (e.g., lower 10% of the distribution).

Circularized assemblies were produced using both the long- and short-read data by Unicycler (v.0.4.7) (37). Following assembly, we ran additional rounds of Pilon (v.1.23) until no further changes were found. Assemblies were annotated using Prokka (v.1.13.3) (38). Structural variants were identified using a combination of Sniffles (v.1.0.12) and Assemblytics (v.1.0). Structural variants were further filtered to high-confidence calls by inspecting the BAM file and filtering to discard minor allele variants (AF of <0.9). Single nucleotide variants were called against the hybrid assemblies using Snippy (v.4.3.6).

**Transcriptome sequencing.** RNA sequencing was performed on biological triplicates of the ancestral and transconjugant strains. For each replicate, a colony was picked from overnight growth on nutrient agar and added to 5 ml of nutrient broth (Sigma-Aldrich, United Kingdom). Cultures were incubated at 37°C until they reached an optical density at 600 nm ($OD_{600}$) of ~0.6. RNA was extracted using TRIzol (ThermoFisher, United Kingdom). Following isopropanol precipitation, DNA was digested using Turbo DNase (ThermoFisher, United Kingdom) based on the manufacturer's protocol. The final RNA solution was purified using a RNeasy minikit (Qiagen, United Kingdom) and quantified using a Qubit with the RNA HS assay kit. The RNA was immediately stored at −80°C.

RNA sequencing was performed by the Centre for Genomics Research (Liverpool, United Kingdom). The RNA integrity number and library insert size were verified using the Agilent RNA 6000 Pico kit and Bioanalyzer platform (Agilent, USA). The RiboZero kit (Illumina, USA) was used to deplete rRNA, and dual indexed libraries were prepared using the NEBNext Ultra Directional RNA sequencing kit (New England Biolabs, USA). Libraries were sequenced on a HiSeq 4000 (Illumina, USA) configured to $2 \times 150$-bp cycles. In order to obtain at least 10 million reads per sample, the sequencing run was distributed across three lanes.

Kallisto (v.0.46.0) was used to quantify differential gene expression, with the high-quality hybrid *de novo* assemblies of parental strains used as a reference. Input files were prepared using Prokka (v.1.13.3) for annotation, genbank_to_kallisto.py (https://github.com/AnnaSyme/genbank_to_kallisto.py) to convert the annotation files for use with Kallisto, and GNU-Parallel (v.20180922) for job parallelization. Differential gene expression was analyzed using Voom/Limma in Degust (v.3.20), with further processing of the resulting differential counts in R (v.3.5.3). UPGMA (unweighted pair group method using average linkages) clustering was performed using DendroUPGMA (http://genomes.urv.cat/UPGMA/). Functional categories were assigned to genes using eggnog-mapper (v.2).

**Plasmid persistence.** pLL35 encodes two TA systems and therefore is likely to be highly stable, but this has not been previously measured. To quantify the persistence of the plasmid over time, 4 replicate cultures of each plasmid-containing strain were propagated by daily serial transfer in nutrient broth (NB) microcosms (6 ml of NB in a 30-ml glass universal vial) for 16 days. One percent of each culture was transferred to fresh medium every 24 h. Populations were plated out onto nutrient agar ± 4 μg/ml of cefotaxime at days 0, 1, 2, 4, 7, 10, 13, and 16.

**Growth curves and phenotypic profiling.** MIC assays for the parental and transconjugant strains were conducted according to the CLSI guidelines (39), using nutrient broth and cefotaxime. Shaking (180 rpm) overnight cultures in 6 ml nutrient broth were established from independent colonies previously grown on agar plates. The following day, 0.5 McFarland cell suspensions were prepared and further diluted 1/500 to inoculate 200 μl of nutrient broth in 96-well plates. The final cefotaxime concentrations tested were 0.25, 0.5, 0.75, 1, 1.25, 1.5, and 1.75, followed by 2-fold increases from 2 to 2,048 μg/ml, and the final volume per well was 200 μl (100 μl bacterial inoculum plus 100 μl antibiotic solution). The $OD_{600}$ was recorded after 24 h of static incubation at 37°C and normalized by subtracting the $OD_{600}$ of a blank well. As working with positive $OD_{600}$ values facilitates further data analysis and interpretation, we linearly transformed $OD_{600}$ estimates by adding 0.0938876 to all data. For each strain and plasmid combination, the relative growth at each antibiotic concentration was obtained by dividing $OD_{600}$ values in the presence of antibiotic by the $OD_{600}$ of the corresponding parental strain grown in the absence of antibiotic. The relative growth values were used to calculate the area under the curve (AUC) with the auc function from the R package flux (flux_0.3-0). Statistical analyses were performed on Box Cox transformed data to fulfil ANOVA assumptions.

Growth kinetics for both plasmid-free and plasmid-containing strains were measured using cultures grown in 200 μl nutrient broth per well in a 96-well plate using an automated absorbance plate reader as a single assay containing all replicates (Tecan Spark 10). Wells were inoculated using the same procedure described above. Plates were incubated at 37°C for 39 h, and the optical density of each well was measured every 30 min at 600 nm. Plates were shaken for 5 s (orbital shaking, movement amplitude of 3 mm, 180 rpm) and allowed to settle for 50 s prior to each reading. A humidity cassette was used to minimize evaporation of the samples.

**Data availability.** Sequence data are available under BioProject no. PRJNA667580, and individual SRA accession numbers are provided in Table S1.

## SUPPLEMENTAL MATERIAL

Supplemental material is available online only.
**FIG S1**, PDF file, 1 MB.
**FIG S2**, PDF file, 0.3 MB.
**FIG S3**, PDF file, 0.01 MB.
**FIG S4**, PDF file, 0.4 MB.
**FIG S5**, JPG file, 0.1 MB.
**FIG S6**, PDF file, 2.2 MB.
**FIG S7**, PDF file, 0.7 MB.
**TABLE S1**, XLS file, 0.03 MB.
**TABLE S2**, XLS file, 0.02 MB.

## ACKNOWLEDGMENT

This work was funded by a BBSRC project grant (BB/R006261/1 and BB/R006253/1) jointly awarded to A.M. and M.B., respectively.

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
