## [Reviewer comments · mSystems]

Limited and strain-specific transcriptional and growth responses to acquisition of a multidrug resistance plasmid in genetically diverse *Escherichia coli* lineages

Steven Dunn, Laura Carrilero, Michael Brockhurst, and Alan McNally

Corresponding Author(s): Alan McNally, University of Birmingham

Review Timeline:

Submission Date:	January 22, 2021
Editorial Decision:	March 22, 2021
Revision Received:	April 1, 2021
Accepted:	April 2, 2021

Editor: David Cleary

Reviewer(s): Disclosure of reviewer identity is with reference to reviewer comments included in decision letter(s). The following individuals involved in review of your submission have agreed to reveal their identity: David A. Baltrus (Reviewer #2)

Transaction Report:

DOI: <https://doi.org/10.1128/mSystems.00083-21>

March 22, 2021

Prof. Alan McNally
University of Birmingham
Birmingham
United Kingdom

Re: mSystems00083-21 (Limited and strain-specific transcriptional and growth responses to acquisition of a multidrug resistance plasmid in genetically diverse Escherichia coli lineages)

Dear Prof. Alan McNally:

I have received reviews for your manuscript and both reviewers were very positive about your manuscript with only some minor comments/revisions to address.

Below you will find the comments of the reviewers.

To submit your modified manuscript, log onto the eJP submission site at <https://msystems.msubmit.net/cgi-bin/main.plex>. If you cannot remember your password, click the "Can't remember your password?" link and follow the instructions on the screen. Go to Author Tasks and click the appropriate manuscript title to begin the resubmission process. The information that you entered when you first submitted the paper will be displayed. Please update the information as necessary. Provide (1) point-by-point responses to the issues raised by the reviewers as file type "Response to Reviewers," not in your cover letter, and (2) a PDF file that indicates the changes from the original submission (by highlighting or underlining the changes) as file type "Marked Up Manuscript - For Review Only."

Due to the SARS-CoV-2 pandemic, our typical 60 day deadline for revisions will not be applied. I hope that you will be able to submit a revised manuscript soon, but want to reassure you that the journal will be flexible in terms of timing, particularly if experimental revisions are needed. When you are ready to resubmit, please know that our staff and Editors are working remotely and handling submissions without delay. If you do not wish to modify the manuscript and prefer to submit it to another journal, please notify me of your decision immediately so that the manuscript may be formally withdrawn from consideration by mSystems.

Sincerely,

David Cleary

Editor, mSystems

Journals Department
Reviewer comments:

Reviewer #1 (Comments for the Author):

In this work, Dunn and colleagues investigated the transcriptional and fitness effects of an antibiotic resistance plasmid of clinical relevance in a collection of eight wild type *E. coli* isolates. Results revealed that this plasmid produced subtle (and uncorrelated) transcriptional and growth effects in the host bacteria. This work presents interesting novelties compared to previous studies on the transcriptional/fitness effects of plasmids. First, the antibiotic resistance plasmid is a clinically important *incF* plasmid encoding the CTX-M-15 ESBL, and the bacterial hosts include *E. coli* ST131, which are usually associated with similar plasmids and represent the most common cause of extra-intestinal *E. coli* infections. In addition, the authors sequenced the bacterial genomes before and after conjugation, which is an important and useful control. Finally, this is one of the few works on plasmid biology combining fitness assays and transcriptomic analyses. In summary, although there are certain limitations in the interpretation of the results due to the subtle effects produced by the plasmids, I think this work will represent a valuable and significant contribution to the field.

Comments

It would be interesting to test if the differences in cefotaxime resistance levels in the plasmid carrying strains are associated with differences in the expression levels of the plasmid-encoded blaCTX-M-15 gene in these strains. It is known that the resistance level conferred by beta-lactamase-type enzymes increases linearly with the amount of enzyme. Could the authors use the transcriptomic data to look into this?

The small number of significantly differently transcribed genes could have to do with potential variation among biological replicates of RNA extractions. Could the authors show how the different replicates of RNA extractions of each strain group in a PCA-type analysis (basically like in figure S6, but showing each independent biological replicate for each plasmid-carrying and plasmid-free isolates).

Just a suggestion: although relative growth parameters extracted from growth curves are a good proxy for fitness, competition assays provide higher sensitivity detecting small changes in relative fitness. This approach could provide a more accurate analysis of the correlation between the transcriptomic changes and plasmid fitness effects.

Line 328. Isn't a >2-fold log change in expression (>4x) a bit too stringent as a threshold? In figure 3 there seem to be many more genes with a significant difference in expression that may be interesting to include in the fitness/transcriptomic changes correlation.

Reviewer #2 (Comments for the Author):

The manuscript from Dunn et al. describes characterization of phenotypic responses and plasmid dynamics across a range of *E. coli* isolates upon acquisition of plasmid pLL35. This work follows up on a few earlier papers from different taxa, demonstrating that plasmid acquisition can have unique effects depending on strain backgrounds, and seeks to match up these effects with transcriptional responses.

Overall I think the experiments are well done, include appropriate controls, are interpreted judiciously, and are sufficient to form a published paper. As such I really only have a few editorial comments to make:

L29: "we observed significant transcriptional..." it strikes me that this result is in the abstract but it kind of buried in the paper, just weird to me to emphasize here

L41: better as "by transferring adaptive functions between lineages that often enable invasion of new niches, including"

L45: delete "more ecologically relevant". I'd just expand this a bit and make the case that natural isolates are relevant because they can be reservoirs that enable maintenance and transmission of plasmids to clinically relevant strains.

L49: better as "specific effects on bacterial growth and transcription under laboratory conditions"

L64: delete "strains"

L94: there's gotta be a better word than "homeostasis" to use here

L99: "preadapted" strikes me as a loaded term. Maybe something like "more tolerant to potential disruptions from plasmid acquisition"

L147: best to spell out what the "phenotypes indicative of *E. coli*" are here

L156: "basal" better as "confounding effects of"

L163: please provide details about how strains were prepped for genome sequencing (how passaged, etc...). Please also explicitly say if same DNA was used for short and long read sequencing

L175: DNA sheared at all?

L262: please provide details about replication on the 96 well plates. Single assay with experimental reps? Multiple assays?

On behalf of all authors, I would like to offer our sincere gratitude to the editor and reviewers for their time and appraisal of our manuscript. It is refreshing to be in receipt of such constructive and helpful comments. We have incorporated the changes suggested by reviewers, and have taken their excellent suggestions on board for future works. We have provided a point by point response to reviewer comments below:

Reviewer #1

It would be interesting to test if the differences in cefotaxime resistance levels in the plasmid carrying strains are associated with differences in the expression levels of the plasmid-encoded blaCTX-M-15 gene in these strains. It is known that the resistance level conferred by beta-lactamase-type enzymes increases linearly with the amount of enzyme. Could the authors use the transcriptomic data to look into this?

This is an excellent point, and something we tried to investigate. We looked at expression levels of plasmid genes, normalised to the counts assigned to the Rep gene. This showed a highly consistent level of transcription across all plasmid genes, in all isolates. We also looked at plasmid copy number as a potential mechanism for potentiation of CTX-M, however this also was extremely consistent across all samples. We are conducting further research into this, but do not believe we can answer this question using the data contained within this manuscript

The small number of significantly differently transcribed genes could have to do with potential variation among biological replicates of RNA extractions. Could the authors show how the different replicates of RNA extractions of each strain group in a PCA-type analysis (basically like in figure S6, but showing each independent biological replicate for each plasmid-carrying and plasmid-free isolates).

We have included supplemental figure 7 containing PCA plots of individual replicates, and agree that variation amongst the replicates could have had an impact on the number of variants reaching FDR-significance.

Just a suggestion: although relative growth parameters extracted from growth curves are a good proxy for fitness, competition assays provide higher sensitivity detecting small changes in relative fitness. This approach could provide a more accurate analysis of the correlation between the transcriptomic changes and plasmid fitness effects.

We agree that competition assays would be a more sensitive way to assess fitness effects of plasmid acquisition. However, we are working here with natural isolates and do not have the necessary labelled strains in all of these backgrounds to perform competition experiments. We attempted to use a small plasmid encoding a fluorescence protein but this was unstable and highly costly, and therefore unsuitable for the purpose of competition experiments. We note that a recent similar study showed a high degree of correlation between fitness estimates of plasmid costs obtained by competition assay versus growth assay suggesting that our growth assays are informative albeit not the gold-standard (San Millán, 10.1101/2020.08.01.230672).

Line 328. Isn't a >2-fold log change in expression (>4x) a bit too stringent as a threshold? In figure 3 there seem to be many more genes with a significant difference in expression that may be interesting to include in the fitness/transcriptomic changes correlation.

Only a relatively small number of genes met FDR significance and of these most displayed high log fold changes. As such, using a stringent 2 fold log change threshold best represents the significant changes in expression we observed. Note however that we do also go on to discuss genes with expression changes that occurred below this threshold in the results and discussion (lines 350-380) and provide plots of these (Figs 3, 5, S5).

Reviewer #2

L29: "we observed significant transcriptional..." it strikes me that this result is in the abstract but it kind of buried in the paper, just weird to me to emphasize here

This line describes a whole section of the Results (i.e. subheading: Functions whose expression was affected by pLL35 varied between strains") thus we do not agree that it is odd to highlight this as a key finding. We have removed the term significant as this may be as over emphasizing the point being made.

L41: better as "by transferring adaptive functions between lineages that often enable invasion of new niches, including"

We have reworded this sentence.

L45: delete "more ecologically relevant". I'd just expand this a bit and make the case that natural isolates are relevant because they can be reservoirs that enable maintenance and transmission of plasmids to clinically relevant strains.

We agree that this needed more work, and have expanded the sentence to include the reviewer's excellent suggestions.

L49: better as "specific effects on bacterial growth and transcription under laboratory conditions"

We have added 'under laboratory conditions'.

L64: delete "strains"

Thank you for spotting this error – it has been removed.

L94: there's gotta be a better word than "homeostasis" to use here

We agree that homeostasis doesn't fit in this context, and have changed it to [disruption of] 'normal cellular function'.

L99: "preadapted" strikes me as a loaded term. Maybe something like "more tolerant to potential disruptions from plasmid acquisition"

We have reworded this sentence in line with the reviewer's suggestion, and have removed the word preadapted.

L147: best to spell out what the "phenotypes indicative of E. coli" are here

We now describe the phenotype as pink coloured colonies.

L156: "basal" better as "confounding effects of"

We have replaced 'basal' with 'confounding effects of'

L163: please provide details about how strains were prepped for genome sequencing (how passaged, etc...). Please also explicitly say if same DNA was used for short and long read sequencing

Independent DNA preps from overnight suspensions were used for each technology. Illumina sequenced DNA was prepared by MicrobesNG, who performed their own SPRI based extraction using a robotic liquid handler. We have included these details to the Methods section, along with the existing description of a phenol chloroform method for MinION extraction. We hope this is now clearer.

L175: DNA sheared at all?

We did not shear the DNA – in our experience under the standard LSK109 protocol the SPRI clean-ups offer sufficient shearing of phenol/chloroform prepped DNA. We have referenced the manufacturer's protocol, and mentioned that the DNA was unsheared.

L262: please provide details about replication on the 96 well plates. Single assay with experimental reps? Multiple assays?

We have included detail specifying a single assay with associated replicates per plate.

April 2, 2021

Prof. Alan McNally
University of Birmingham
Birmingham
United Kingdom

Re: mSystems00083-21R1 (Limited and strain-specific transcriptional and growth responses to acquisition of a multidrug resistance plasmid in genetically diverse Escherichia coli lineages)

Dear Prof. Alan McNally:

Thank you for submitting your revised manuscript. I am satisfied that you have addressed all the reviewers comments and therefore I am very pleased to tell you that your manuscript has been accepted and I am forwarding it to the ASM Journals Department for publication. For your reference, ASM Journals' address is given below. Before it can be scheduled for publication, your manuscript will be checked by the mSystems senior production editor, Ellie Ghatineh, to make sure that all elements meet the technical requirements for publication. She will contact you if anything needs to be revised before copyediting and production can begin. Otherwise, you will be notified when your proofs are ready to be viewed.

- Minimum resolution of 1280 x 720
- .mov or .mp4. video format
- Provide video in the highest quality possible, but do not exceed 1080p
- Provide a still/profile picture that is 640 (w) x 720 (h) max

We recognize that the video files can become quite large, and so to avoid quality loss ASM suggests sending the video file via <https://www.wetransfer.com/>. When you have a final version of the video and the still ready to share, please send it to Ellie Ghatineh at eghatineh@asmusa.org.

Thank you for submitting your paper to mSystems.

Sincerely,

Supplemental Material: Accept
Supplemental Material: Accept
Supplemental Material: Accept
Supplemental Material: Accept
Supplemental Material: Accept
Supplemental Material: Accept
Supplemental Material: Accept
Supplemental Material: Accept
Supplemental Material: Accept